# 3D Printing Technology Based on Versatile Gelatin-Carrageenan Gel System for Drug Formulations

**DOI:** 10.3390/pharmaceutics15041218

**Published:** 2023-04-11

**Authors:** En Liang, Zengming Wang, Xiang Li, Shanshan Wang, Xiaolu Han, Daquan Chen, Aiping Zheng

**Affiliations:** 1Collaborative Innovation Center of Advanced Drug Delivery System and Biotech Drugs, School of Pharmacy, Yantai University, Yantai 264005, China; 2Beijing Institute of Pharmacology and Toxicology, 27th Taiping Road, Haidian District, Beijing 100850, China; wangzm.1986@163.com (Z.W.);

**Keywords:** 3D printing, pediatric medicines, printability, rheological characteristics, microstructure

## Abstract

Currently, there is a shortage of pediatric medicines on the market, and 3D printing technology can more flexibly produce personalized medicines to meet individual needs. The study developed a child-friendly composite gel ink (carrageenan-gelatin), created 3D models by computer-aided design technology, then produced personalized medicines using 3D printing to improve the safety and accuracy of medication for pediatric patients. An in-depth understanding of the printability of different formulations was obtained by analyzing the rheological and textural properties of different gel inks and observing the microstructure of different gel inks, which guided the formulation optimization. Through formulation optimization, the printability and thermal stability of gel ink were improved, and F6 formulation (carrageenan: 0.65%; gelatin: 12%) was selected as the 3D printing inks. Additionally, a personalized dose linear model was established with the F6 formulation for the production of 3D printed personalized tablets. Moreover, the dissolution tests showed that the 3D printed tablets were able to dissolve more than 85% within 30 min and had similar dissolution profiles to the commercially available tablets. This study demonstrates that 3D printing is an effective manufacturing technique that allows for flexible, rapid, and automated production of personalized formulations.

## 1. Introduction

There are still many shortcomings in pediatric medicine, such as a lack of drug varieties, appropriate dosage forms, and specifications which affect the normal use of pediatric medicine [1]. Because children are not “scaled down” adults, their bodies and organs are immature in terms of drug tolerance, dosage, and administration. The traditional way of dosing by splitting up the pills is one of the shortcomings in pediatric medicine which can lead to an inaccurate and unsafe dose for children [2]. To address these issues, the subdivision of pediatric medicine is commonly done using temporary dispensing, oral solutions, oral liquid, pill-splitter, mini-tablets, automatic powder subcontracting machines, and 3D printing technology [3,4,5]. In particular, 3D printing offers a new approach to pharmaceuticals, breathing fresh life into pediatric preparations requiring customized dosages and combinations.

Semi-solid extrusion (SSE) is one of the 3D printing technologies based on gel or paste inks being deposited layer-by-layer to build three-dimensional structures [6]. Over the past few years, several studies have been conducted using SSE 3D printing for personalized medicines development. Goyanes et al. had shown that 3D printing can be an effective manufacturing technology for producing chewable isoleucine “printlets”, with good acceptability as a treatment for maple syrup urine disease in 2019 [7]. Zheng et al. had shown that 3D printing technology can prepare spironolactone and hydrochlorothiazide split-dose tablets by grinding commercial tablets and adding excipients to prepare ink, which proved to be a possible new method for hospital drug formulation [8]. Olmos et al. had found that 3D printed ink-based on sodium alginate-cellulose nanofibers, filled with drugs of different solubility (curcumin and chloramphenicol), enabled on-demand release of personalized tablets [9]. In addition, SSE 3D printing technology can be used to produce bilayer controlled-release tablets [10], gastric floating tablets [11], multiple release tablets [12], etc.

The in-depth study of SSE 3D printing has shown that paste inks require time consuming post-processing such as freeze drying, vacuum drying, etc., and require 12 h or more for solvent evaporation [12,13]. In contrast, the gelation mechanism of gel inks is mainly physical, chemical, or radiation crosslinks and can be cured rapidly even at low or room temperature without post-processing. For example, Azam et al. had showed that 3D printing technology could be used to produce personalized vitamin gummies with the components of gum arabic, guar gum, carrageenan, and xanthan [14], and the formulation could be cured and molded without post-processing. Tagami et al. had found that a hydrogel composed of lamotrigine, gelatin, hydroxypropyl methylcellulose, reduced syrup, and water could be used to produce tablets of different shapes and colors by 3D printing, with no post-processing of the formulation [15]. In recent years, gel inks have become a promising approach as SSE 3D printing inks have been extensively investigated in the fields of oral and topical drug delivery, tissue regeneration and biomedicine, and food [16].

However, for the single-gel system as a 3D printing ink, its mechanical properties or material stability is poor, resulting in low structural stability and accuracy of printing targets as well as poor printing results. Gelatin is a natural polymeric peptide polymer produced by moderate hydrolysis of collagen, due to its low cost and good gelling, solubility, biocompatibility, emulsification, and other advantages. To improve the poor mechanical properties of single-gelatin matrices, better performing gel materials can be obtained by chemical modification and/or coupling with other molecules. For example, Valentina et al. evaluated the biophysical properties of an innovative photo cross-linked hydrogel based on gelatin-methacryloyl, suggesting that it may be a promising new biomaterial with potential applications in cartilage regeneration [17]. Gelatin can form a thermally reversible gel with water, exhibiting a gel state at low temperatures (below 25 °C) and a solution state at high temperatures (above 35 °C) [18]. This property may limit its use in pharmaceutical foods due to its shortcomings in thermal stability. To solve the problem of thermal stability of gelatin, polysaccharides such as carrageenan [19], gellan gum [20], or starch-based polymers [21] can be added to improve the thermal stability of gelatin. Carrageenan is a polysaccharide with thermal responsiveness and gelling ability that can improve the thermal stability and mechanical strength of the gel ink when combined with gelatin [22].

In this study, SSE 3D printing technology was used to investigate a formulation with propranolol hydrochloride as an active pharmaceutical ingredient for personalized medicine study in children to meet the personalized dose requirements of children with arrhythmia. The study systematically analyzed the effects of rheological properties, textural properties, and microstructure of gel inks on the appearance and accuracy of printed tablets, and a gel ink with good printability and stability was screened. Also, by controlling the number of layers of printed tablets in the research, different doses of tablets are possible, and doctors can be more flexible in meeting the individual needs of their patients. In addition, 3D printed chewable tablets are equivalent or superior to traditional commercial tablets in terms of appearance, dissolution, taste, compliance, and other aspects. Moreover, the process control strategy for the expanded production of gel ink was preliminarily explored, and it was possible to realize the batch production of gel ink and ensure the continuous supply of ink. It was shown that SSE 3D printing technology has the capability of small-lot production and can be more flexible to meet the dispensing needs of pharmacies. 

## 2. Materials and Methods

### 2.1. Materials

Propranolol hydrochloride (Changzhou Tianhua Pharmaceutical Co., Ltd., Changzhou, Jiangsu, China); propranolol hydrochloride tablets (Jiangsu Yabang Epsom Pharmaceutical Co., Ltd., Shanghai, China); gelatin (Anhui Fengyuan Gelatin Co., Ltd., Bengbu, Anhui, China); carrageenan (Fujian Green Power Biotechnology Co., Ltd., Zhangzhou, Fujian, China); carboxymethyl starch sodium (CMS-Na) (Huzhou Prospect Co., Ltd., Huzhou, Zhejiang, China); glucose (Weifang Shengtai Pharmaceutical Co. Ltd., Weifang, Shandong, China); water (Watson’s Food & Beverage Guangzhou, Guangzhou, Guangdong, China); methanol, acetonitrile (Thermo Fisher Scientific Co., Ltd., Waltham, Massachusetts, USA); hydrochloric acid, phosphoric acid (Sinopharm Chemical Reagent Co., Ltd., Shanghai, China).

### 2.2. Gel Inks Preparation

The amount of carrageenan and CMS-Na was investigated for the printability of gel inks. The compositions of F1, F2, F3, F4, F5, F6, and F7 are shown in Table 1. (1) Gelatin swelling and dissolution: A quantity of gelatin and pure water was weighed and allowed to swell in a beaker at room temperature for 30 min. The swollen gelatin was then placed in a drum of a planetary mixer (2 L planetary mixer; Mixer Stirring & Kneader Co., Ltd., Xiangtan, Hunan, China) at 50 °C for 20 min. (2) Mixing: The remaining ingredients were added to the drum and mixed at 80 °C for 10 min. (3) Ink filling: The prepared gel inks were filled into a 20 mL printer-compatible syringe (Inject^®^ Luer Solo, B. Braun Medical Inc. Bethlehem, Germany) and into sealed polyester (PET) bags for storage.

### 2.3. Rheological Test

Rheological properties of carrageenan-gelatin mixtures were measured with rheometer (Discovery HR-10, TA Instruments, New Castle, DE, USA) with a 20 mm plate at a gap of 1 mm. Prior to the test, the sample was heated to a flowing state at 60 °C. To simulate the rheological properties of the 3D printing process, the print temperature was chosen as the test temperature.

The viscosity tests were performed with a shear rate ranging from 0.1 to 100 s^−1^ in order to obtain the viscosity change variation of inks. This experiment uses the generalized power-law viscosity model, where n is the flow index and indicates Newtonian behavior when n = 1; n < 1 and n > 1 indicate shear thinning and thickening [23], respectively (Equation (1)).
η = Κ ∙ γ^(n−1)^(1)
where η stands for viscosity (Pa·s, γ represents the shear rate (s^−1^), k represents the power-law consistency index (Pa·s), and n refers to the power-law index).

As for the strain, sweeps were carried out in a strain range of 0.1 to 1000% at a fixed frequency of 1 Hz to determine the linear viscosity range (LVR) of inks. The shear recovery tests were used to characterize the shear recoverability of gel inks by applying a low shear rate of 1 s^−1^ for 180 s, followed by a high shear rate of 100 s^−1^ for 20 s, and, finally, at a low shear rate of 1 s^−1^ for 180 s. The temperature sweep tests were also performed at a fixed frequency (1 Hz) with a temperature fall step from 90 °C to 25 °C and a temperature rise step from 25 °C to 90 °C at a rate of 4 °C/min. These three experiments were performed to understand the temperature sensitive, supportability, and the shift of the modulus with the temperature of the gel ink.

### 2.4. Texture Profile Analysis of Gel Inks

The textural properties of different gel inks were analyzed by using a texturizer (TA XTC, Shanghai Baosheng Industrial Development Co., Ltd., Shanghai, China). Texture profile analysis (TPA) measures textural properties by contacting a cylindrical probe with a gel block (a block obtained by filling a mound of the same size with gel ink). The TPA test is a comprehensive reflection of the hardness, chewiness, springiness, adhesiveness, cohesiveness, and resilience of an object by means of the secondary downward pressure of the probe. The test speed was 1 mm/s with a trigger force of 5 gf, and the return speed after the test was 2 mm/s.

### 2.5. Optical Microscopy and Scanning Electron Microscopy

The freeze-dried ink was placed on a glass slide and the different formulations were observed at 10× magnification using a light microscope (BX53 Light Microscope, Olympus Corporation, Tokyo, Japan) and photographed for preservation. The microstructures of different gel inks and the 3D printed objects by F6 formulation were measured using a scanning electron microscope (SEM) (S-4800, Hitachi Limited, Tokyo, Japan) at an accelerating voltage of 5.0 kV. All samples were lyophilized and then tested.

### 2.6. Printability Assessment and Tablet Printing

A disposable syringe containing gel ink was placed in the hot chamber of the 3D printer and heated at the print temperature for 20 min. As shown in Figure 1, the design model is imported into the printing software and sliced after the printing parameters are set so that the printability assessment and tablet preparation of the gel ink can be performed. The equipment used for the experiment was a drug 3D printer (M3DIMAKER™, Fab Rx Ltd., London, UK). The print speed was 25 mm/s, and the nozzle diameter was 0.8 mm.

Test 1: A hollow cylindrical shape (diameter: 15 mm, height: 30 mm) was printed and the strength and printability of the 3D structures printed from each gel ink were observed and assessed in terms of defects and collapse. Test 2: The circular tablets (diameter: 12.5 mm, height: 5 mm, filling rate was 15%) were printed and their weight variation was calculated (should be no more than 5%, 21 tablets/plate) to evaluate the accuracy of the gel inks. The same set of slice codes was used for each ink printability study.

### 2.7. Model Design and Dose of 3D Printed Personalized Tablets

The F6 formulation was used to establish the relationship between personalized dosing and model parameters for 3D printing. In this study, when the formulation was fixed, the assay was of a certain value and the dose of the 3D printed chewable tablets was determined by weight. The circular-shaped tablets model (diameter: 20.3 mm, height: 0.8 * h mm, where h was the number of printed layers, and the fill rate was 15%) allows different doses (1 mg, 2 mg, 3 mg, 4 mg, and 5 mg) in the range 1 to 5 mg to be dispensed by printing different layers of chewable tablets. The nozzle diameter was 0.8 mm, the print speed was 25 mm/s, and the layer height was 0.8 mm. The number of print preparations was all 21 tablets/plate, and the weight variation of tablets should be no more than 7.5% (1 mg to 3 mg) or 5% (3 mg to 5 mg).

### 2.8. Dissolution Test

The dissolution test was performed to use USP35 dissolution tester II (RC1207DP, Tianda Tianfa Technology Co., Ltd., Tianjin, China). The tests were performed at a paddle speed of 75 rpm using 1000 mL of dilute hydrochloric acid (1 in 100) as the dissolution medium at a temperature of 37 ± 0.5 °C. The dissolution test was performed to use commercially available preparations and 3D printed preparations (dose 1: 2 mg, dose 2: 5 mg), by cutting test preparations (dose: 5 mg) to simulate chewing. Samples of 5 mL were withdrawn at 5, 10, 15, 20, 30, and 45 min, and the same volume of dissolution medium was added simultaneously. The concentration of the drug was determined by the content method and the average dissolution rate was calculated.

## 3. Results and Discussion

### 3.1. Rheological

#### 3.1.1. Viscosity Test

The semisolid that can be used as an ink depends on the rheological properties of the material. During the extrusion stage of the gel ink, pseudoplasticity is essential to ensure that the gel ink is extruded from the narrow nozzle. As shown in Figure 2A, the shear viscosity test results for the seven formulations F1 to F7 show a rapid decrease in apparent viscosity with increasing shear rate, indicating that all of them are pseudoplastic fluids with shear thinning properties that facilitate extrusion of gel inks from the nozzle. Gel inks for 3D printing should be selected with the appropriate n, k, and yield stress to ensure extrudability and good mechanical properties. The shear thinning viscosity of all formulations according to the rheological test results shown in Table 2 with n < 1 and a fit index of 0.9782 or more. The k increased from 193.45 Pa·s (F1) to 403.35 Pa·s (F4), demonstrating that the addition of carrageenan was able to increase the viscosity of gel inks. The k also increased from 193.45 Pa·s (F1) to 1016.67 Pa·s (F2), indicating that the viscosity of the gel ink increased significantly with the addition of CMS-Na. The F6 formulation showed that the addition of carrageenan and CMS-Na synergistically increased the viscosity of inks (Table 2 and Figure 2A). The intersection of storage modulus (G′) and loss modulus (G″) is defined as the yield stress by recording G′ and G″ in the oscillatory stress sweep [24]. The F1 formulation had the lowest yield stress (69.89 Pa) when compared to the remaining formulations, and its poor support led to the failure of the print (Table 2). This shows that the composite gel ink has a significant increase in viscosity and yield stress compared to the single gel system, which improves the printability of inks. The yield stress of F3 and F7 formulations were 1568.54 Pa and 1448.57 Pa, respectively. Both formulations have excellent mechanical and anti-deformation properties, requiring higher extrusion forces and temperatures.

#### 3.1.2. Strain Sweep

Strain sweep tests provide an understanding of the rheological properties of gel inks such as the yield stress, G′ and G″. As shown in Figure 2B, the gel ink exhibited solid-like properties (G′ > G″) with no change in storage modulus over the strain range of 0.1–10% for the seven formulations, indicating that the gel ink underwent reversible deformation at the LVR. This indicates that the network structure of the gel ink was irreversibly deformed, and the structure was disrupted when the gel ink exceeded the LVR. As shown in Figure 2B, the mechanical strength of the gel ink was significantly improved by adding carrageenan and CMS-Na to the formulation. It was shown that the mechanical strength of the composite gel ink (e.g., F6) was better than that of the single gel ink (e.g., F1).

#### 3.1.3. Shear Recovery Behavior

In the extrusion process of SSE 3D printing, the ink is squeezed in the printer and nozzle, resulting in a large deformation. Thus, the ink must have sufficient mechanical strength to support the layer-by-layer printing of the object, which requires the ink to have good thermalization to quickly return to a stable state. As shown in Figure 3A, the viscosity of the F1 formulation was lower than the other formulations and did not reach a stable viscosity as the viscosity increased with shear time at the same shear rate. In addition, its poor mechanical properties made it difficult to print the desired objects. Figure 3A,B shows that the viscosity of the F2 formulation was higher than F1 and it took 23 s to return to stable viscosity after a high shear rate. Therefore, the printability of the F2 formulation is better than that of F1. In Figure 3A,B, the viscosity of the F4 formulation increased with the addition of carrageenan. It reached a steady viscosity after 4 s of high shear rate, while the difference from the initial viscosity was significant and the thixotropy was not good. As shown in Figure 3A,B, the F6 formulation has a suitable viscosity with a recovery time of 9 s to stable viscosity after a high shear rate, which ensures fast bonding and stacking of filaments during the printing process. CMS-Na and carrageenan work synergistically to increase the viscosity and thixotropy of the gel ink, thereby improving printing performance. It is worth noting that too much carrageenan or CMS-Na will cause the gel ink to become poorly flowable, which is detrimental to the production and filling of gel inks and ultimately affects the accuracy of 3D printed objects (e.g., F3).

#### 3.1.4. Temperature Sweep Test

Gel inks used in SSE 3D printing are mainly viscoelastic semisolid, allowing them to exhibit both solid and fluid properties. Temperature sweep cooling tests of F1, F2, F4, F5, and F6 formulations are shown in Figure 4A with initial G′ < G″ and slowly increasing G′ and G″ as the temperature decreased. Over a range of temperatures, the G′ and G″ of the gel ink increased rapidly, with G′ eventually becoming greater than G″, indicating that as the temperature decreased, the elastic properties began to dominate and the transition from sol to gel occurred. As shown in Figure 4A,B, the gelation temperature of the F1 gel ink was at 25 °C. When CMS-Na was added, the gelation temperature of F2 gel ink increased to 34 °C. When carrageenan was added to the F4 gel ink, the gelation temperature increased significantly to 52 °C. It has been shown that CMS-Na and carrageenan could increase the gelation temperature and improve the temperature sensitivity of a single gelatin gel system. Nevertheless, as shown in Figure 4A,B, G′ decreased with cooling of F3 and F7 formulation and modulus decreased significantly in the temperature range from 60 °C to 50 °C. Both G′ were greater than G″ and the gel ink exhibited mainly solid-like properties with no sol-gel transition behavior in the temperature range tested.

It is shown in Figure 4C,D that each gel ink for F1, F2, F4, F5, and F6 started with G′ > G″, and both G′ and G″ gradually decreased, indicating a gradual collapse of the gel network structure. Above a certain temperature, the modulus of the gel ink drops sharply. The cross of G′ and G″ represents a ‘gel-sol’ phase transition. As shown in Figure 4C, the melting temperature of the F1 gel ink was at 40 °C. After the addition of CMS-Na, the melting temperature of the F2 gel ink increased to 48 °C. As shown in Figure 4D, after the addition of carrageenan to the gel ink of the F4 formulation, the melting temperature was increased to 76 °C. For the F3 and F7 formulations, G′ became larger than G′ with increasing temperature, showing solid-like properties, and there was no intersection between G′ and G″, indicating that no gel-sol transition occurred in the temperature range tested (Figure 4C,D).

Overall, the rheological properties of the F1 formulation suggested that it was not suitable for 3D printing. Therefore, it was necessary to add excipients such as carrageenan and CMS-Na to improve the printability. As shown in Figure 4E,F, the addition of carrageenan and CMS-Na synergistically increased the complex modulus (G*) of the formulation and improved the supportability and printability of the gel ink. The addition of carrageenan to the formulation increased the “sol-gel” phase temperature of the gel system and the acceptable range of printing temperatures, improving the thermal sensitivity of the single-gelatin system, and contributing to the accuracy of the formulation printing. In summary, the F6 formulation met the printing requirements with the right viscosity, good support properties, and high printing accuracy.

### 3.2. Texture Profile Analysis of Gel Inks

The textural properties of seven formulations were measured in the TPA test mode. As shown in Figure 5A,B, the hardness and chewiness of the F2 and F4 formulations were significantly improved compared to the F1 formulation, indicating that carrageenan and CMS-Na enhanced the textural properties of gel inks and improved their hardness and chewiness. As shown in Figure 5A,B, the hardness and chewiness of the F2 and F4 formulations were significantly improved compared to the F1 formulation. These results indicated that carrageenan and CMS-Na improved the textural properties of gel inks and increased their hardness and chewiness. The results of the F4, F5, F6, and F7 formulation tests showed that the textural property of gel inks increased and then decreased as the amount of CMS-Na increased. (Figure 5A,B). The addition of more CMS-Na allows more starch molecules to be dispersed into the spaces between the gelatin molecules, resulting in a more solid gel ink. However, when the formulation was added at 9% (F7), the starch molecules interfered with the interaction of the gel molecules, resulting in poor textural properties of the gel ink such as hardness and gumminess. In conclusion, the F6 formulation performed better in terms of hardness, etc. (Detailed data of Figure 5B are shown in Appendix A).

### 3.3. Microstructure Characterization and Printing of Gel Inks

The microstructures of different formulations were observed by optical microscopy and SEM, as shown in Table 3. As the F1 and F4 formulations did not contain CMS-Na, their microscopic characterization showed that the surface was smooth and dense. In addition, the filament cannot be extruded from the nozzle in a homogeneous manner, so the printability is poor and does not meet the requirements of printing. The microstructures of F2, F3, F5, F6, and F7 are shown in Table 3. The inclusion of CMS-Na in the formulation results in a more compact and consistent extruded filament structure, which improves the support properties and printability of the gel ink. Due to the unbranched structure of the carrageenan macromolecule and its polyanionic nature, it can form a high-viscosity colloidal solution, which can affect the flow of the liquids. Similarly, adding too much CMS-Na can reduce the fluidity of the gel ink.

As shown in the microstructure of the extruded filaments of the F1 formulation in Table 3, the filaments were not homogeneous and smooth, along with their low viscosity and poor support properties, made it impossible to print three-dimensional structural objects. The addition of carrageenan improved both viscosity and mechanical properties (e.g., F4). However, this was not sufficient to support the deposition of each layer. Based on the analysis of the rheological, textural, and microstructural properties of the gels, the inclusion of CMS-Na and carrageenan in the formulation could synergistically improve the stacking and formation properties of the gels. CMS-Na was the key to ensuring the formability of the gel ink. Comparing the printing of each formulation in Table 3 and analyzing the printing appearance of the formulations as well as the difference in tablet weight (printing accuracy), the printing results of F5, F6, and F7 formulations were better. Furthermore, when considering the processes used to manufacture and fill the gel inks, the F6 formulation is considered to be optimal.

### 3.4. Microscopic Characterization of Tablet Structures

The surface and internal structure of the chewable tablets made with F6 gel ink were observed by SEM, as shown in Figure 6A,D. The tablets showed a grid-filled internal structure with uniform filaments, indicating that the gel ink has good support properties and is resistance to deformation. This formulation enables the printing of personalized preparations with good appearance and high accuracy for better application to patients.

### 3.5. Model Design and Dose of 3D Printed Personalized Tablets

In this study, process parameters were optimized to enable the production of different dosage preparations by printing different tablet layers based on the controlled diameter of the cylindrical tablet model. As shown in Table 4, the upper and lower weight limits for each size of the preparation were calculated by calculating the individual weights of each 3D printed tablet; all weights met the criteria. This study established a linear equation between the dose of the 3D printed personalized chewable tablets and the theoretical number of layers: y = 185.95 x + 36.783 with a correlation coefficient r = 0.9998, which is a good fit. This showed that with a good formulation and printing process parameters, personalized dose adjustment can be achieved by controlling the number of printed layers on the model.

### 3.6. Dissolution Result

Dissolution tests were used to simulate the in vitro dissolution behavior of the preparation and to predict drug bioavailability and potency of the drug. Dissolution profiles were studied for both sizes of the test preparation and the commercially available preparation, and all released more than 85% of the drug within 30 min (shown in Figure 7) and the dissolution behavior of the test preparation was similar to that of the commercially available preparation. The dissolution profile was studied by cutting the test preparation into pieces to simulate patient chewing, and by testing the whole preparation to simulate accidental swallowing. Due to the porous network structure of the 3D printed chewable tablets, more than 85% of the drug was released within 30 min, regardless of whether the tablets were cut or not, meeting the drug dissolution criteria. This means that even if the child accidentally swallows the tablet while taking it, the release of the drug is unaffected.

## 4. Conclusions

This study used SSE 3D printing technology to develop a thermally-stable gelatin-carrageenan composite gel ink using propranolol hydrochloride as a model drug, which allowed flexible dose regulation of pediatric medicines and provided a new idea for personalized medicine tailoring. First, the study designed gelatin-carrageenan composite gel systems with different ratios to improve the printability and the thermal stability issues of the single-gel ink. Second, the printing properties of different gel inks can be better understood by analyzing the rheological and textural properties and observing the microstructure of different gel inks. Third, the formulation was determined based on the printability and print accuracy of the different formulations. The personalized dose model was investigated using SSE 3D printing technology based on the optimal formulation F6. The 3D printing process was optimized based on the fixed diameter of the cylindrical tablet model, which allowed the weight of each layer to be within an acceptable range. The regulation of the personalized dose was achieved by printing different layers. A linear equation was established between the theoretical number of layers and the drug dose, and the results showed a good fit, with correlation coefficients in line with the requirements. Finally, the dissolution profiles of the test preparations were investigated, and the 3D printed chewable propranolol hydrochloride tablets were found to be superior or equivalent to conventional commercial preparations in terms of dissolution, dosage accuracy, acceptability of appearance, and so on.

SSE 3D printing technology has the advantage of being flexible, fast, and ready to be manufactured at room temperature, and has been used in exploratory applications for dose-splitting in both national and international hospital or community pharmacy settings. The drug-loaded gelatin-carrageenan composite gel ink has good printability and stability is good for SSE printing technology. At the same time, the small-batch production of personalized preparations is realized with the help of a good printing process. This will be followed by formulation stability studies and drug crystal structure studies to support the development of tailored pediatric medicines. In the future, it is believed that 3D printing technology will better serve personalized medicine through appropriate control strategies and the development of online quality control systems, as well as the continuous improvement of relevant laws and regulations led by regulatory authorities.

## Figures and Tables

**Figure 1 pharmaceutics-15-01218-f001:**
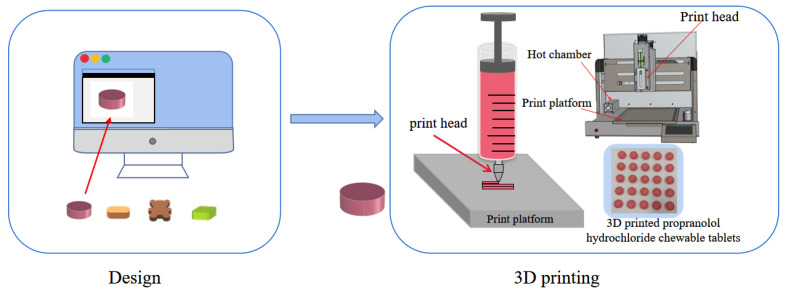
Image of the manufacturing process of 3D printed products.

**Figure 2 pharmaceutics-15-01218-f002:**
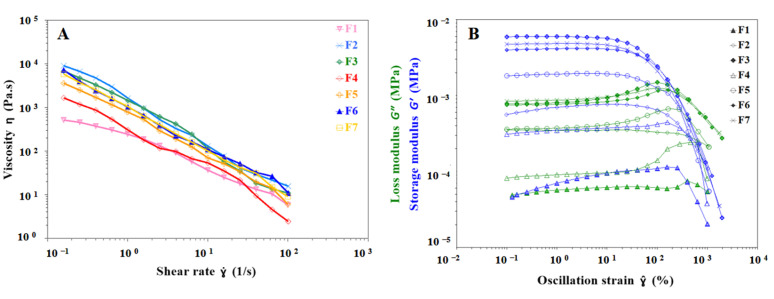
Images of viscosity test and strain sweep results for different gel inks. (**A**) Viscosity behavior of different formulations at shear rates from 0.1 s^−1^ to 100 s^−1^. (**B**) Strain sweep test results for different formulations to determine the linear viscoelastic region of gel inks.

**Figure 3 pharmaceutics-15-01218-f003:**
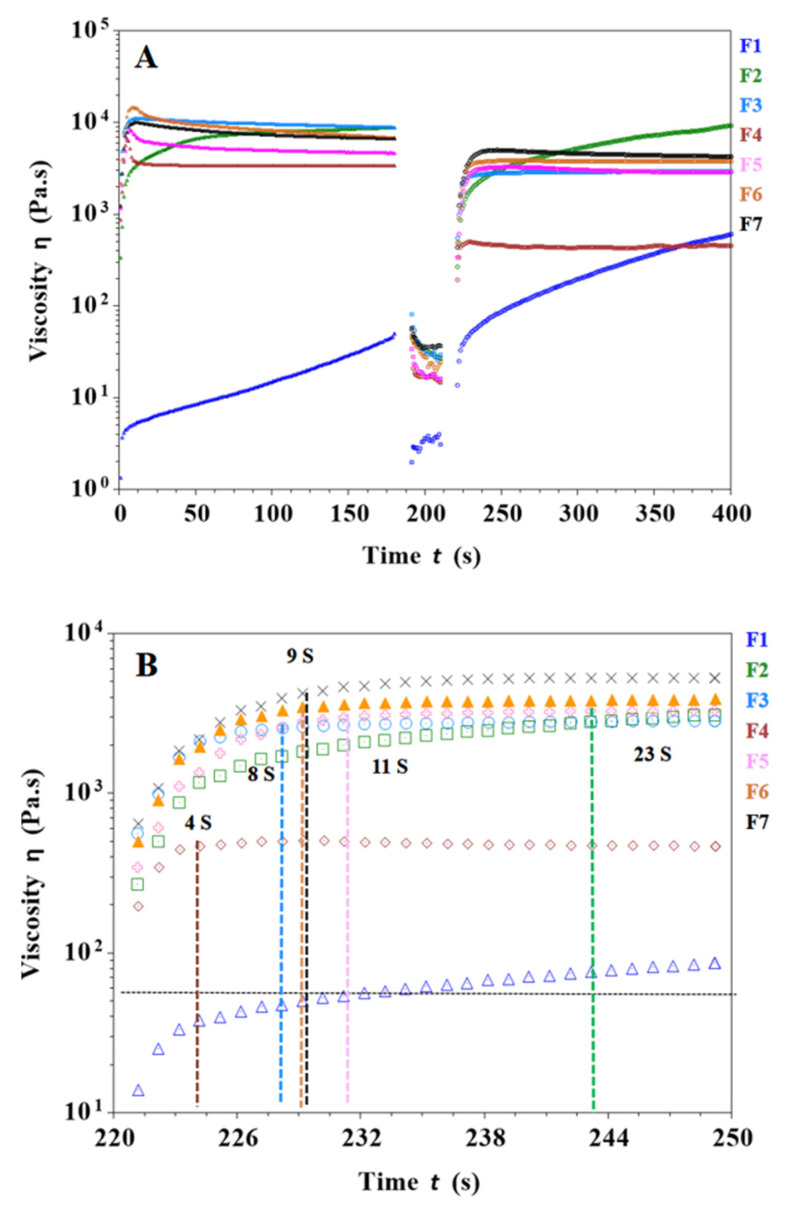
Results of shear recovery tests for different gel inks. (**A**) Thixotropy tests of different gel inks at low (1 s^−1^) and high (100 s^−1^) shear rate changes. (**B**) Recovery of viscosity in the third step (at 221st second begin) for different formulations to determine the time required to recover stable viscosity for different gel inks.

**Figure 4 pharmaceutics-15-01218-f004:**
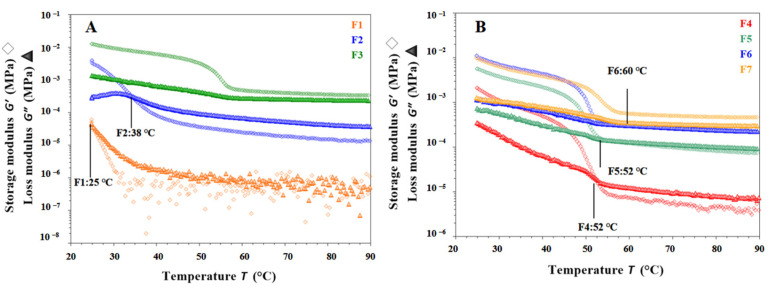
Result of temperature sweep tests for different gel inks. (**A**,**B**) The changes in G′ and G″ during cooling (90 °C to 25 °C) for different gel inks. (**C**,**D**) The changes in G′ and G″ during heating (25 °C to 90 °C) for different gel inks. (**E**) The changes in G* during cooling for different gel inks. (**F**) The change in G* for different formulations during temperature rise.

**Figure 5 pharmaceutics-15-01218-f005:**
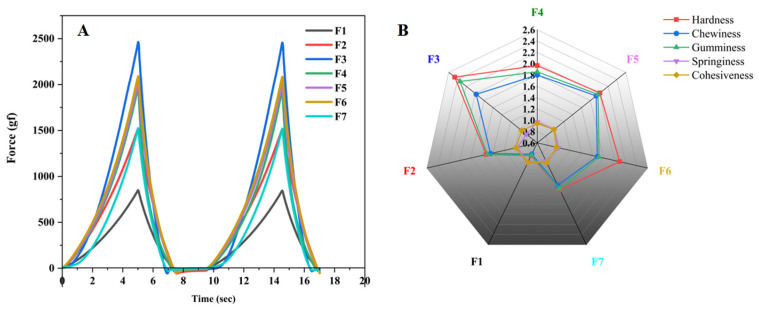
Images of the TPA test result. (**A**) TPA curve of each formulation. (**B**) Radar chart of texture characteristics of each formulation.

**Figure 6 pharmaceutics-15-01218-f006:**
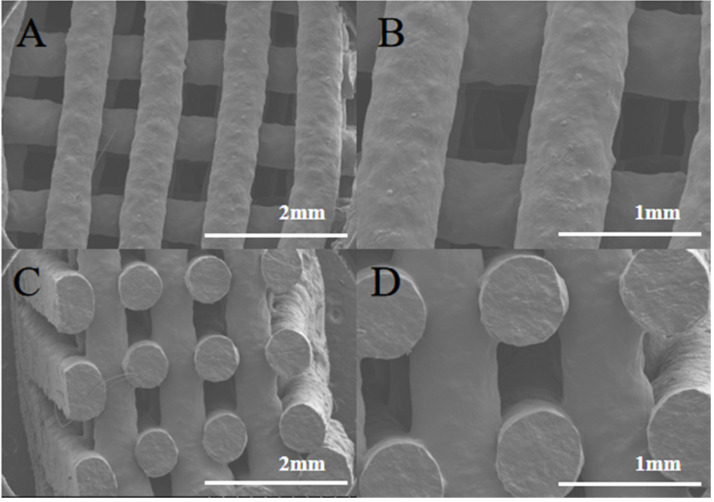
SEM image of chewable tablet structure. (**A**) SEM observation of the surface of the chewable tablet (×25). (**B**) SEM observation of the surface of the chewable tablet (×50). (**C**) SEM observation of the cross section of the chewable tablet (×25). (**D**) SEM observation of the cross section of the chewable tablet (×50).

**Figure 7 pharmaceutics-15-01218-f007:**
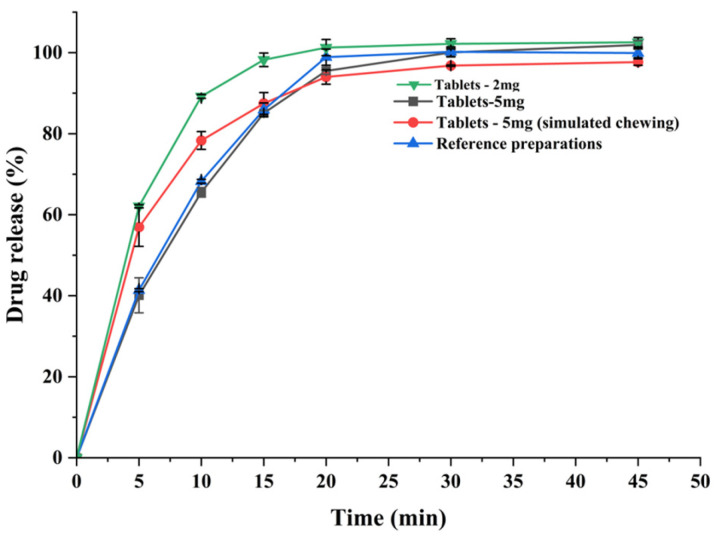
Dissolution curve of the 3D printed chewable tablets and the commercially available reference.

**Table 1 pharmaceutics-15-01218-t001:** The weight percentage of different components in the seven groups of gel inks.

Gel Ink	Propranolol Hydrochloride (%)	Gelatin (%)	Carrageenan (%)	CMS-Na (%)	Others * (%)
F1	0.50	12.00	0.00	0.00	87.50
F2	0.50	12.00	0.00	6.00	81.50
F3	0.50	12.00	1.30	6.00	80.20
F4	0.50	12.00	0.65	0.00	86.85
F5	0.50	12.00	0.65	3.00	83.85
F6	0.50	12.00	0.65	6.00	80.85
F7	0.50	12.00	0.65	9.00	77.85

* Others include water (40%), glucose, pigments.

**Table 2 pharmaceutics-15-01218-t002:** Parameters of rheological properties of gel inks.

Gel Ink	K (Pa·s)	n	Yield Stress (Pa)	R^2^
F1	193.45 ± 42.89	0.1695	69.89 ± 2.63	0.9782
F2	1016.67 ± 155.68	−0.2188	325.94 ± 53.07	0.9790
F3	1367.68 ± 216.27	−0.0642	1568.54 ± 277.50	0.9904
F4	403.35 ± 66.67	0.0830	248.92 ± 26.72	0.9977
F5	701.82 ± 65.88	−0.0211	691.23 ± 137.78	0.9933
F6	1109.81 ± 45.03	0.1163	993.48 ± 6.32	0.9969
F7	1197.42 ± 196.48	0.1330	1448.57 ± 147.75	0.9990

**Table 3 pharmaceutics-15-01218-t003:** Table of microstructure characteristics and printability assessment results of different gel inks.

Gel Ink	Print Temperature (°C)	Optical Microscopy	SEM	3D Structure Evaluation	Appearance	Weight Variation (%)
Gel Ink	Gel Ink	Filament	Top	Side	Bottom
F1	36	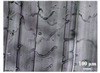	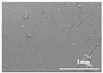	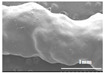	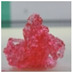	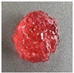	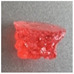	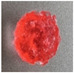	±8.1
F2	37	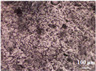	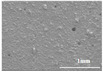	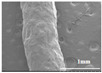	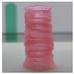	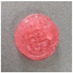	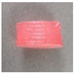	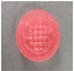	±12.8
F3	42	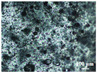	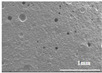	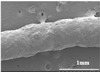	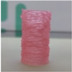	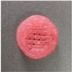	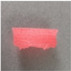	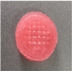	±8.0
F4	38	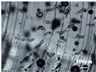	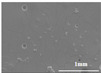	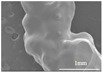	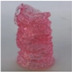	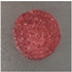	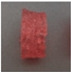	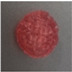	±8.7
F5	39	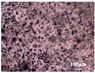	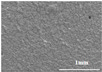	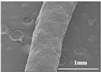	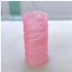	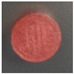	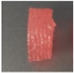	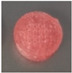	±3.1
F6	40	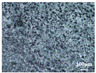	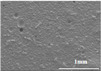	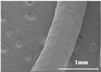	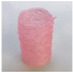	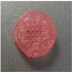	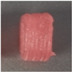	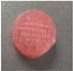	±3.5
F7	41	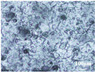	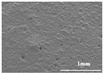	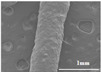	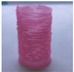	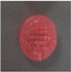	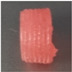	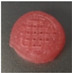	±3.3

**Table 4 pharmaceutics-15-01218-t004:** 3D printed model of a personalized chewable tablet and print result table.

Dose Range (mg)	Layers	Model Size (mm)	Mean Weight (g) ± SD	Weight Compliance Limits (g)	r (Linear Fitting Equation: y = 185.95 x + 36.783)
1.00	1	20.30 × 20.30 × 0.80	0.2263 ± 0.0064	0.2093–0.2433	0.9998
2.00	2	20.30 × 20.30 × 1.60	0.4083 ± 0.0049	0.3879–0.4287	
3.00	3	20.30 × 20.30 × 2.40	0.5884 ± 0.0057	0.5590–0.6178	
4.00	4	20.30 × 20.30 × 3.20	0.7797 ± 0.0076	0.7407–0.8187	
5.00	5	20.30 × 20.30 × 4.00	0.9770 ± 0.0058	0.9219–1.0189	

## Data Availability

The data presented in this study are available on request from the corresponding author. The data are not publicly available due to some privacy issues about drug development.

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
