# Peer review of "3D Printing Technology Based on Versatile Gelatin-Carrageenan Gel System for Drug Formulations"

_pharmaceutics, 2023, doi:10.3390/pharmaceutics15041218_

Round 1

Reviewer 1 Report

In the article: “3D printing technology based on versatile gelatin-carrageenan gel system for drug formulations”, the authors discussed about the development of a child-friendly composite gel ink (carrageenan-gelatin) to  produce personalized medicines using 3D printing to improve the safety and accuracy of medication for pediatric patients.

Overall, this manuscript results very interesting, the authors clearly explain the rational of the study and discussed the topic point by point.

However, we would like to invite the authors  to clarify some minor points:

 1.      Please check the check punctuation and spaces;

2.   Why the choice of gelatin as starting materials? Please, within the introduction section describe and  explain in general the properties of this molecule;

3.    Among the introduction, the authors described the poor properties of sole gelatin, it should be chemically modified and/or coupled to other molecules. Please deepen this topic and also describe the demonstrated biological activity of gelatin, in this  respect the following reference should be useful: Vassallo V, Tsianaka A, Alessio N, Grübel J, Cammarota M, Tovar GEM, Southan A, Schiraldi C. Evaluation of novel biomaterials for cartilage regeneration based on gelatin methacryloyl interpenetrated with extractive chondroitin sulfate or unsulfated biotechnological chondroitin. J Biomed Mater Res A. 2022 Jun;110(6):1210-1223. doi: 10.1002/jbm.a.37364. Epub 2022 Jan 28. PMID: 35088923; PMCID: PMC9306773.

4.       Did the authors check the biocompatibility of the obtained formulations? It is possible perform a standard test, such as MTT assay?

5.      Why the use of Propranolol hydrochloride in the formulation?

Author Response

Point 1: Please check the check punctuation and spaces
Response 1: Thank you for your suggestion. We have checked and corrected the
punctuation and spaces throughout the text.

Point 2: Why the choice of gelatin as starting materials? Please, within the introduction section describe and explain in general the properties of this molecule
Response 2: Thank you for your suggestion. The reason we chose gelatin as the raw material is that gelatin is a natural polymeric peptide polymer produced by moderate hydrolysis of collagen, which has good gelling properties and low cost, and is a good gelling agent. We have added this part in the introduction, line 74-76.

Point 3: Among the introduction, the authors described the poor properties of sole
gelatin, it should be chemically modified and/or coupled to other molecules. Please deepen this topic and also describe the demonstrated biological activity of gelatin, in this respect the following reference should be useful: Vassallo V, Tsianaka A, Alessio N, Grübel J, Cammarota M, Tovar GEM, Southan A, Schiraldi C. Evaluation of novel biomaterials for cartilage regeneration based on gelatin methacryloyl interpenetrated with extractive chondroitin sulfate or unsulfated biotechnological chondroitin. J Biomed Mater Res A. 2022 Jun;110(6):1210-1223. doi: 10.1002/jbm.a.37364. Epub 2022 Jan 28. PMID: 35088923; PMCID: PMC9306773.
Response 3: Thank you for your suggestion. We have added a description in the
introduction. It is necessary for our article to deepen how to improve the properties of single gelatin, and we have added this part in the introduction, line76-78.

Point 4: Did the authors check the biocompatibility of the obtained formulations? It is possible perform a standard test, such as MTT assay?
Response 4: Thank you for your suggestion. The excipients we use are all approved
pharmaceutical excipients, such as gelatin, carboxymethyl starch and glucose, which are commonly used in tablet manufacturing. And the active ingredients are marketed and approved. The amounts of excipients are all below FDA safe doses and no potentially toxic ingredients have been added. Next, to ensure the effect of the excipient components on the active ingredient propranolol hydrochloride, we will consider conducting raw material and excipient compatibility testing at a later stage, but this part is for the drug declaration and is not very relevant to this article, so it is not considered in the design of the article. Thank you for your understanding.

Point 5: Why the use of Propranolol hydrochloride in the formulation?
Response 5: Thank you for your suggestion. Propranolol hydrochloride is the main
active ingredient in the printing ink, and in this project, we conducted a study on 3D printing personalized tablets using propranolol hydrochloride as a model drug for the clinical split-dose requirement of cardiac arrhythmia in children. From 2016 to 2019, the Chinese government has developed three batches to encourage research and development to declare drugs for children, and propranolol hydrochloride is in the first batch of the list. That is why we have chosen propranolol hydrochloride as the active ingredient.

Reviewer 2 Report

In  the current article the authors have developed 3D printed paediatric dosage forms using gelatine carrageenan inks. The work is very interesting an the authors have provide a full methodology of how to prepare such printed structures. They conducted a range of rheological studies to evaluate the suitability of the gels.

The discussion is well structured and justified by the obtained experimental results. I have two minor queries before the article is publishable:

- why the authors didn't attempt to print at room temperatures by adjusting the ratio of the excipients?

-please add in detail the protocol for the evaluation of chewiness, gumminess and springiness as they are missing from the Methods.

Author Response

Point 1: Why the authors didn't attempt to print at room temperatures by adjusting the ratio of the excipients?

Response 1: Thanks for your suggestion. It is possible to print at room temperature by adjusting the ratio of excipients, but objects printed at room temperature may require post-processing processes such as vacuum drying or freeze drying. In addition, our research is aimed at developing a personalized preparation with good thermal stability to facilitate storage and transport of the tablets and to prevent the tablets from melting under excessive storage conditions or in hot weather, which would affect patient safety.

Point 2: Please add in detail the protocol for the evaluation of chewiness, gumminess and springiness as they are missing from the Methods.

Response 2: Thanks for your suggestion. We have added data source descriptions of chewiness, gumminess and springiness to the methodology. We have added this at line 167-170 of the method.
